# Post-Hypoglycemic hyperglycemia are highly relevant markers for stratification of glycemic variability and partial remission status of pediatric patients with new-onset type 1 diabetes

**Antoine A. Harvengt**[1,2☯], **Olivier G. Polle**[1,2☯], **Manon Martin**[3], **Aline van Maanen**[4],
**Laurent Gatto**[3], **Philippe A. Lysy**[1,2]*

1 Pôle PEDI, Institut de Recherche Expérimentale et Clinique, UCLouvain, Brussels, Belgium, 2 Specialized Pediatrics Service, Cliniques universitaires Saint-Luc, Brussels, Belgium, 3 Computational Biology and Bioinformatics (CBIO) Unit, de Duve Institute, UCLouvain, Brussels, Belgium, 4 Statistical Support Unit, Institut Roi Albert II, Cliniques universitaires Saint-Luc, Brussels, Belgium

☯ These authors contributed equally to this work.

* philippe.lysy@uclouvain.be

**Data Availability Statement:** The clinical data used in this study cannot be publicly released due to

## Abstract

### Aims

To evaluate whether parameters of post-hypoglycemic hyperglycemia (PHH) correlated with glucose homeostasis during the first year after type 1 diabetes onset and helped to distinguish pediatric patients undergoing partial remission or not.

### Methods

In the GLUREDIA (GLUcagon Response to hypoglycemia in children and adolescents with new-onset type 1 DIAbetes) study, longitudinal values of clinical parameters, continuous glucose monitoring metrics and residual β-cell secretion from children with new-onset type 1 diabetes were analyzed during the first year after disease onset. PHH parameters were calculated using an in-house algorithm. Correlations between PHH parameters (i.e., PHH frequency, PHH duration, PHH area under the curve [PHH$_{AUC}$]) and glycemic homeostasis markers were studied using adjusted mixed-effects models.

### Results

PHH parameters were strong markers to differentiate remitters from non-remitters with PHH/Hyperglycemia duration ratio being the most sensitive (ratio<0.02; sensitivity = 86% and specificity = 68%). PHH$_{AUC}$ moderately correlated with parameters of glucose homeostasis including TIR ($R^2$ = 0.35, p-value < 0.05), coefficient of variation ($R^2$ = 0.22, p-value < 0.05) and Insulin-Dose Adjusted A1c (IDAA$_{1C}$) ($R^2$ = 0.32, p-value < 0.05) and with residual β-cell secretion ($R^2$ = 0.17, p-value < 0.05). Classification of patients into four previously described glucotypes independently validated PHH parameters as reliable markers of

ethical and legal considerations. Researchers interested in accessing these data are requested to contact the UCLouvain Legal and Intellectual Property Department. To obtain access, a data transfer contract must be drawn up in accordance with the procedures available at the following address: https://uclouvain.be/fr/repertoires/entites/rjur.

**Funding:** Grant funding from Leona M. and Harry B. Helmsley Charitable Trust The funders had no role in study design, data collection and analysis, decision to publish, or preparation of the manuscript.

**Competing interests:** The authors have declared that no competing interests exist.

glucose homeostasis and improved the segregation of patients with intermediate values of $IDAA_{1C}$ and estimated C-peptide ($CPEP_{EST}$). Finally, a combination of PHH parameters identified groups of patients with specific patterns of hypoglycemia.

## Conclusion

PHH parameters are new minimal-invasive markers to discriminate remitters from non-remitters and evaluate glycemic homeostasis during the first year of type 1 diabetes. PHH parameters may also allow patient-targeted therapeutic management of hypoglycemic episodes.

## Introduction

Type 1 diabetes mellitus is characterized by a progressive decline in β-cell mass, resulting in a clinical state of insulinopenia when β-cell function drops below twenty percent [1], which defines the clinical disease onset. From that moment, patients with type 1 diabetes rely on a combination of exogenous insulin administration, healthy diet and regular physical activity to achieve optimal glycemic control [2–4]. Immediately after disease onset and insulin therapy initiation, the majority of patients (occurrence rate: [40–75%]) experience a period of partial remission defined by the coexistence of low levels of glycemic variability and reduced exogenous insulin requirements [5]. However, after partial remission, a progressively growing dependence on exogenous insulin induces an increased glycemic variability and the difficulty for patients to avoid hypoglycemia. Indeed, despite major improvements in diabetes management [6], nearly half of type 1 diabetes patients do not reach recommended therapeutic targets [7]. Also, under classical intensive insulin therapy, an incompressible hypoglycemia frequency (estimated at 5–15% of total glucose values [8]) is unavoidable to maintain mean glycemia within targets.

Hypoglycemia is the most common complication in type 1 diabetes patients [8] and corresponds to an inadequacy between insulin substitution, insulin needs and carbohydrate intake with a consecutive drop of glycemia (i.e., below 60 mg/dL) [9]. Hypoglycemic events are either asymptomatic or associated with mild-to-severe clinical manifestations, such as convulsions and/or loss of consciousness [10,11]. Though rarely life-threatening [8], children experience on average three symptomatic hypoglycemic events per week that commonly require external intervention (i.e., providing oral carbohydrates or glucagon analogs) [8]. Additionally, as hypoglycemia is an acute, stressful and unpleasant event for both the child and the parents, these events may lead to excessive carbohydrate intake followed by acute hyperglycemia and, more globally, increased glycemic variability [12]. This highlights hypoglycemia as a potential trigger of hyperglycemic excursions and the need for therapeutic education focusing on individual profiles of glycemic variation.

In patients with type 1 diabetes, hypoglycemic events were associated with increased oxidative stress [13–15] that participates in the development of microvascular complications (e.g., diabetic retinopathy and nephropathy) [16,17]. This was clinically demonstrated by Ceriello et al. who observed increased expression of oxidative stress markers in patients experiencing hypoglycemia followed by hyperglycemia, while the same markers remained unchanged when a normoglycemic state was maintained [18]. These results suggest a link between inadequate carbohydrate intake during hypoglycemia and the development of diabetes complications.

Based on these observations, our team recently introduced the concept of post- hypoglycemic hyperglycemia (PHH) and investigated their influence on diabetes control (**E**valuation of

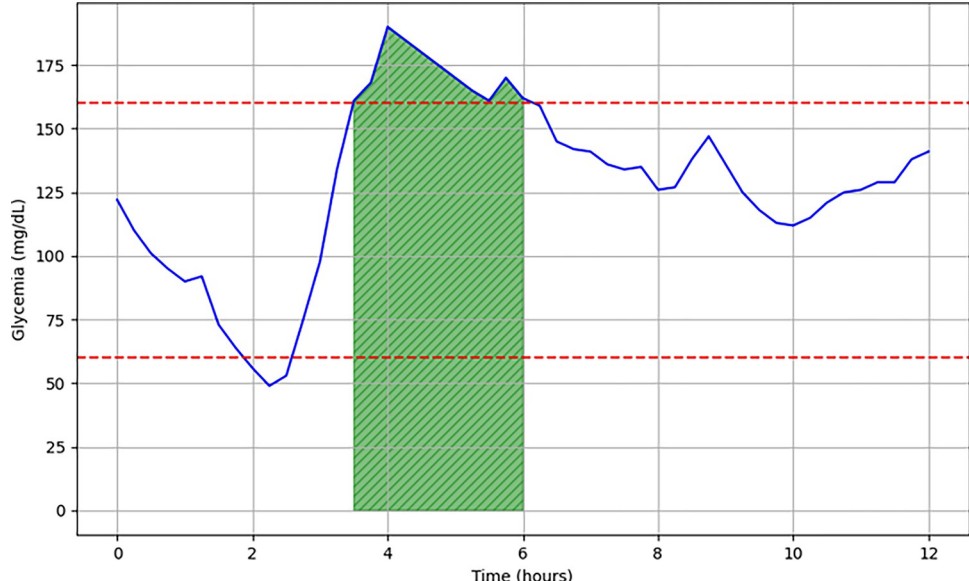

**Fig 1. Graphical representation of a continuous analysis of subcutaneous glucose (vertical axis) over a period of 12 hours (horizontal axis).** The normoglycemia range is 60–160 mg/dL. In this example: End of hypoglycemia at 02:15 p.m., onset of PHH at 03:15 p.m., end of PHH at 06:00 p.m. Hyperglycemia occurred less than 2 hours after the end of hypoglycemia; PHH lasted for 2 hours and 45 minutes. The green area is the PHH AUC.

**P**osthypoglycemic **H**yperglycemia **i**n **C**hildren and **A**dolescents with diabetes study, EPHICA study) [19]. PHH was defined as hyperglycemia (i.e., glycemia >160 mg/dL), that is preceded, within a 2-hour period, by hypoglycemia (i.e., glycemia <60 mg/dL) (**Fig 1**). In pediatric patients with longstanding type 1 diabetes (i.e., clinical onset >1 year), these glycemic patterns represented more than a third of time spent in hyperglycemia for nearly 15% of patients and displayed strong correlations with markers of glycemic variability [19]. While current β-cell function markers lack reliability in reflecting the extent of glycemic homeostasis in patients with type 1 diabetes, our recent results support PHH as reliable and minimal-invasive markers of glycemic homeostasis [20] that may help individualize therapeutic interventions.

The objectives of our GLUREDIA study were to characterize PHH events in a cohort of pediatric patients with new-onset type 1 diabetes and investigate whether PHH parameters correlated with glycemic homeostasis including clinical parameters (e.g., Glycated hemoglobin [$HbA_{1C}$], total daily dose of insulin [TDD], Insulin-Dose Adjusted $A_{1C}$ [$IDAA_{1C}$]), continuous glucose monitoring metrics or residual β-cell secretion (i.e., estimated c-peptide [$CPEP_{EST}$]). Also, we evaluated whether these PHH parameters supported the clinical heterogeneity suggested by the newly identified glucotypes [21] during the first year after type 1 diabetes onset.

## Methodology

### Study design & participants

GLUREDIA is a prospective study that includes data from the multicentric DIATAG (DIAbetes TAGging)study that was previously described [21]. Briefly, DIATAG included new-onset pediatric patients with type 1 diabetes aged between 6 months and 18 years old. Type 1 diabetes was diagnosed using International Society for Pediatric and Adolescent Diabetes (ISPAD) criteria [1] and patients were positive for at least one anti-islet autoantibody (i.e., anti-insulin, anti- protein tyrosine phosphatase, anti-glutamic acid decarboxylase, anti-Zinc transporter 8).

All participants and their parents gave their written consent before enrolment in the study. The protocol was approved by the seven participating ethical committees (Comité d'Ethique Hospitalo-Facultaire of Cliniques universitaires Saint-Luc [CUSL] - 2018/04DEC/462) and is registered in www.clinicaltrial.gov (NCT04007809). Exclusion criteria are described elsewhere [21]. All participants were without significant comorbidities at inclusion (i.e., high blood pressure (>P95), proteinuria (>0.15 g/L), or body mass index [BMI] Z-score >+3 DS according to Cole et al. [22]).

A data array was collected at diagnosis and included demographics of the patient (i.e., age at diagnosis, sex, pubertal status, weight, height and BMI) and diabetes characteristics (i.e., presence of ketoacidosis at diagnosis, anti-islet antibodies, insulin regimen). From diagnosis, clinical parameters (i.e., $IDAA_{1C}$, $HbA_{1C}$, TDD) and raw continuous glucose monitoring metrics were collected at each outpatient clinical visit (i.e., every 3 months) for 1 year. All data were captured using the Research Electronic Data Capture (REDCap) system provided by Vanderbilt University (Nashville, USA) and hosted at CUSL.

### Continuous glucose monitoring metrics analysis and PHH detection

Raw continuous glucose monitoring metrics from a 90-day interval were extracted at each outpatient clinical visit from various continuous glucose monitoring devices (i.e., Freestyle Libre®, Abbott; Dexcom®, Dexcom; EnliteTM, Medtronic MiniMed). Raw glycemic data were pre-processed using the R (R Core Team (2021)) statistical package. Data quality checks and calculation of a panel of forty-six continuous glucose monitoring metrics were performed using the Iglu statistical package.

For PHH detection, raw continuous glucose monitoring metrics were aggregated on T-SQL™ (Azure SQL Instance) and, quality-checked and optimized using an in-house PHH detection algorithm implemented on C# 7.3 using.NET framework. continuous glucose monitoring metrics exhibiting a time-lapse above 45 minutes between consecutive glycemic values were excluded from the analysis. Glycemic values below 20 mg/dL or displaying a glycemic change above 100 mg/dL in less than 5 minutes were considered as artifacts and further excluded from the analysis. Manual inspection of PHH patterns for algorithm accuracy check was performed using PowerBI (Microsoft™). PHH event was defined as a hypoglycemic episode (i.e., interstitial glucose <60 mg/dL) followed, within two hours, by a hyperglycemic episode (i.e., interstitial glucose >160 mg/dL). The PHH event starts when a value above 160 mg/dL is detected within two hours after hypoglycemia. If a glycemic value below 160 mg/dL (e.g., 155 mg/dL) is identified during the PHH event and followed, within a maximum of 15 minutes, by a glycemic value above 160 mg/dL; the PHH event continues. The ends when two consecutive glycemic values below 160 mg/dL (i.e., >15 minutes) are detected. The PHH duration corresponds to the difference in minutes between the start and the end of PHH. The area under the curve of PHH ($PHH_{AUC}$) was calculated using the irregular polygon rule [23] where PHH starts and ends are defined as y = 0. The PHH/Hyperglycemia duration ratio is determined by the proportion of total duration of all PHH according to the total time spent in hyperglycemia and the PHH/Hypoglycemia frequency ratio by the hypoglycemia proportion followed by a PHH. This easy-user friendly software will be added to GitHub repository.

### Circadian rhythm evaluation

To determine the circadian rhythm of PHH, we subdivided the whole day into four distinct periods: morning (5 a.m.; 10 a.m.; $morning_{5-10}$), day (10 a.m.; 4 p.m.; $day_{10-4}$), evening (4 p.m.; 10 p.m.; $evening_{4-10}$) and night (10 p.m.; 5 a.m.; $night_{10-5}$).

## Partial remission

Partial remission was defined by $IDAA_{1C} = HbA_{1C} + (4 \times TDD)$ [24], where a score below 9 defines remitters and a score above 9 defines non-remitters.

## Residual C-peptide secretion

Residual C-peptide secretion ($CPEP_{EST}$) was evaluated at +3 and +12 months after diagnosis. Stimulated C-peptide values were estimated using a mathematical formula described by Wentworth *et al* (Loge [$CPEP_{EST}$ + 1] = 0.317 + 0.00956 × BMI − 0.000159 × duration + 0.710 × FCP − 0.0117 × FPG − 0.0186 × $HbA_{1C}$ − 0.0665 × insulin, where BMI is in kg/m$^2$, duration is in days, Fasting C-Peptide [FCP] is in nmol/l, Fasting plasma glucose [FPG] is in mmol/L, $HbA_{1C}$ is in % and insulin is in IU/kg) [25]. Fasting C-peptide and plasma glucose values were determined at the central laboratory of CUSL for all samples. C-peptide was measured using a two-site chemiluminescence immunoassay (LiaisonXL$^{®}$, Diasorin, France).

## Glucotypes

Our research center has recently identified subgroups of patients with similar glucose profiles by analyzing Continuous Glucose Monitoring (CGM) metrics and clinical parameters [21]. Using unsupervised hierarchical clustering, we identified four distinct clusters of patients with unique glycemic patterns. The glucotypes significantly differed in all clinical parameters and CGM metrics. Time spent in normoglycemia (Time in range; $TIR_{70-180}$) was highest in glucotype 1, but progressively decreased during the daytime in glucotype 2, the whole day in glucotype 3, and showed a net drop-in glucotype 4. Hyperglycemia episodes (Time above the range, $TAR_{>180}$) first appeared during the day in glucotype 2, extended to nighttime in glucotype 3, and peaked across the entire day in glucotype 4. In terms of hypoglycemia, the mean incidence of time spent in hypoglycemia (Time below the range; $TBR_{<70}$) was equivalently high in glucotypes 1 and 3, but $TBR_{<70}$ specifically increased in the early morning in glucotype 1 while remaining stable during the whole day in glucotype 3. Therefore, these glucotypes exhibited specific 24-hour profiles of continuous glucose monitoring metrics and refined the current dichotomic definition of partial remission.

## Statistical analyses

Nearly all statistical analyses were using R (R Core Team [2021]. R: A language and environment for statistical computing. R Foundation for Statistical Computing, Vienna, Austria. URL [https://www.R-project.org/]). In the end, Roc analyses were done using SAS v.9.4 ([https://www.sas.com/en_us/home.html]). The statistical significance level used for all analyses was 0.05. PHH parameters were transformed using Box-Cox transformation when needed. Demographic and clinical data are reported as mean ± SD for continuous variables and as numbers and proportions for categorical variables. Comparisons between groups were performed using Student's t-test, chi-square test and linear regression or their nonparametric equivalent (Mann–Whitney U test and Kruskal–Wallis test, respectively) as appropriate. P-values were adjusted for multiple testing with the Bonferroni procedure [26]. Marginal $R^2$ (coefficient of determination) between PHH parameters and secretion, continuous glucose monitoring metrics and clinical parameters were calculated using generalized linear mixed models [27] with R packages lme4 [28] and lmerTest [29] to take into account multiple measurements from the same patient. Models included the methods as fixed effects and patient identity as a random intercept. Residuals were inspected for normality on Q-Q plots.

## Results

### Clinical and anthropometric characteristics of the GLUREDIA cohort

Seventy-one pediatric patients with new-onset type 1 diabetes were quarterly followed during the first year after diagnosis, corresponding to a total of 244 outpatient visits. Of these visits, 52 were excluded as they did not fulfill our pre-established quality criteria or had missing associated clinical data. Final analysis was performed on 192 clinical visits from 66 patients (i.e., 59 patients at +3, 48 patients at +6, 45 patients at +9 and 40 patients at 12 months after the diagnosis) representing a total of 1900000 interstitial glucose values. The baseline characteristics (i.e., at diagnosis) of the cohort are described in **Table 1**.

**Table 1. Cohort description and distribution of PHH parameters.**

| | Total | Remitters | Non-remitters | p-values |
|---|---|---|---|---|
| | ($n = 194$) | ($n = 108$) | ($n = 86$) | |
| **Phenotypic characteristics** | | | | |
| Age–years (at diagnosis) | $10.7 \pm 3.4$ | $11.3 \pm 3.3$ | $9.8 \pm 3.3$ | $0.05$ [†] |
| Gender–Male no (%) | 95 (50) | 71 (66) | 24 (30) | $7.3e^{-7}$ [‡] |
| Pubertal–no (%) | 98 (52) | 64 (59) | 34 (42) | $0.02$ [‡] |
| **Parameters of glycemic homeostasis** | | | | |
| **Clinical parameters** | | | | |
| $Hb_{A1C}$ –% | $6.6 \pm 1.0$ | $6.0 \pm 0.5$ | $7.3 \pm 0.9$ | $2.09\ e10^{-20}$ [†] |
| $IDAA_{1C}$ | $9.0 \pm 1.6$ | $7.9 \pm 0.7$ | $10.4 \pm 1.2$ | $5.32\ e^{-34}$ [†] |
| **Continuous glucose monitoring metrics** | | | | |
| CV–% | $38.9 \pm 8.6$ | $36.1 \pm 8.8$ | $42.5 \pm 6.5$ | $7.18e^{-8}$ [†] |
| MODD–% | $47.9 \pm 18.6$ | $38.2 \pm 13.5$ | $60.6 \pm 16.5$ | $2.01e^{-18}$ [†] |
| Mean glycemia–mg/dL | $134.5 \pm 30.0$ | $118.5 \pm 17.2$ | $155.7 \pm 30.3$ | $1.83e^{-17}$ [†] |
| $TIR_{70\text{-}180}$ –% | $71.7 \pm 15.0$ | $79.4 \pm 11.1$ | $61.5 \pm 13.4$ | $6.35e^{-18}$ [†] |
| $TBR_{<70}$ –% | $9.5 \pm 7.5$ | $10.8 \pm 8.4$ | $7.6 \pm 5.8$ | $4.60e^{-3}$ [†] |
| $TAR_{>180}$ –% | $19.7 \pm 16.2$ | $10.8 \pm 9.0$ | $31.5 \pm 16$ | $1.1e^{-18}$ [†] |
| **PHH parameters** | | | | |
| Frequency–no/day | $0.19 \pm 0.20$ | $0.15 \pm 0.19$ | $0.25 \pm 0.20$ | $9.05e^{-5}$ [‖] |
| $PHH_{AUC}$ | $32023 \pm 30037$ | $18895 \pm 17387$ | $48220 \pm 34278$ | $6.51e^{-12}$ [‖] |
| PHH duration mean (minutes) | $155 \pm 117$ | $105 \pm 75$ | $217 \pm 130$ | $3.38e^{-11}$ [‖] |
| PHH/hyperglycemia duration ratio–no | $0.04 \pm 0.05$ | $0.02 \pm 0.02$ | $0.07 \pm 0.06$ | $1.43e^{-10}$ [‖] |
| PHH/hypoglycemia frequency ratio–no | $0.35 \pm 1.06$ | $0.28 \pm 1.02$ | $0.44 \pm 1.12$ | $4.71e^{-7}$ [‖] |
| **Circadian rhythm of PHH frequency–no** | | | | |
| $Morning_{5\text{-}10}$ | $0.04 \pm 0.07$ | $0.04 \pm 0.09$ | $0.04 \pm 0.06$ | $1$ [‖] |
| $Day_{10\text{-}4}$ | $0.18 \pm 0.27$ | $0.10 \pm 0.26$ | $0.26 \pm 0.32$ | $1.15e^{-4}$ [‖] |
| $Evening_{4\text{-}10}$ | $0.11 \pm 0.17$ | $0.06 \pm 0.10$ | $0.17 \pm 0.21$ | $1.92e^{-5}$ [‖] |
| $Night_{10\text{-}5}$ | $0.03 \pm 0.05$ | $0.01 \pm 0.02$ | $0.04 \pm 0.06$ | $0.07$ [‖] |

Plus-minus values are means ± SD. Percentages may not total 100 due to rounding. Glycemic homeostasis markers and PHH parameters were evaluated at +3, +6, +9, +12 months after diagnosis. Differences between remitters and non-remitters were considered as significant when p-value was under 0.05. [†] student t-test; [‡] Chi-square; [‖] Linear mixed models using Satterthwaite t-test. PHH, post-hypoglycemia hyperglycemia; $HbA_{1C}$, Glycated hemoglobin; $IDAA_{1C}$, Insulin Dose-Adjusted A1C = $HbA_{1C}$ + 4 x insulin doses/day/kg; CV, coefficient of variation for glucose; MODD, mean of daily differences; $TIR_{70\text{-}180}$, Time in range (70–180 mg/dL); $TBR_{<70}$, Time below the range (<70 mg/dL); $GIIH_{<54}$, Grade 2 hypoglycemia (<54 mg/dL); $TAR_{>180}$, Time above the range (>180 mg/dL); $Hyperglycemia_{>250}$, Hyperglycemia above 250 mg/dL; $PHH_{AUC}$: area under the curve of PHH; PHH/Hyperglycemia duration ratio, ratio between the total duration of PHH and the total duration of hyperglycemia; PHH/Hypoglycemia frequency ratio, ratio between daily frequency of PHH and daily frequency of hypoglycemia; $Morning_{5\text{-}10}$, morning time between 5 am and 10 am; $Day_{10\text{-}4}$, daytime between 10 am and 4 pm, $Evening_{4\text{-}10}$, evening time between 4 pm and 10 pm; $Night_{10\text{-}5}$, nighttime between 10 pm and 5 am.

## PHH pattern is frequent during the first year after type 1 diabetes onset

Of the 194 analyzed continuous glucose monitoring metrics, 174 (90%) exhibited at least one PHH event in the 3 months postdiagnosis. Globally, participants presented 0.19 (±0.20) PHH events/day that lasted on average 155 (±117) minutes corresponding to a mean $PHH_{AUC}$ of 32023 (±30037). The PHH/Hyperglycemia duration ratio was 0.04 (±0.05) and the PHH/Hyperglycemia frequency ratio was 0.26 (±0.18), whereas the PHH/Hypoglycemia frequency ratio was 0.35 (±1.06) meaning that, on average, about a third of hypoglycemia were followed by hyperglycemia in our patient cohort (**Table 1**).

We investigated the circadian rhythm of the PHH, as physiological phenomena or behavioral aspects might influence the occurrence of these events (e.g., Somogyi effect [30], exogenous carbohydrate intake). We observed a higher PHH frequency during the $day_{10-4}$ and $evening_{4-10}$ than in the $morning_{5-10}$ and $night_{10-5}$ ($p<0.001$) (**Table 1**). On the other hand, the $PHH_{AUC}$ increased progressively according to the different daytime periods: it was the highest at $night_{10-5}$ ($p<0.05$).

PHH are thus a common phenomenon in patients with *de novo* type 1 diabetes and demonstrate high inter-patient variability (i.e., high SD) for each global parameter. Evaluation of PHH across the day revealed that these occurred more frequently during the $day_{10-4}$ while being of longer duration during the $night_{10-5}$.

## Partial remission is associated with fewer and lower PHH occurrence

We further investigated whether PHH parameters reflected the occurrence and intensity of partial remission in patients with new-onset type 1 diabetes. Among the 193 glucose records studied, 108 (56%) belonged to patients undergoing partial remission.

Remitters presented fewer PHH events ($p<0.001$) and these were of shorter duration ($p<0.001$) and with smaller AUC ($p<0.001$) than the one observed in non-remitters. Also, remitters exhibited less hypoglycemia followed by a PHH (PHH/Hypoglycemia frequency ratio; $p<0.001$) concurring to a four-time decrease in the percentage of time spent in hyperglycemia due to a PHH (PHH/Hyperglycemia duration ratio; $p<0.001$), as compared to non-remitters (**Table 1**). While the biggest $PHH_{AUC}$ were observed during the $night_{10-5}$ for both groups ($p<0.001$); remitters exhibited smaller $PHH_{AUC}$ compared to non-remitters, regardless of the period of the day. Paradoxically, there was no significant difference in PHH frequency in the $morning_{5-10}$ and during the $night_{10-5}$ between remitters and non-remitters ($p>0.05$), contrasting with the two other daytime periods ($day_{10-4}$ and $evening_{4-10}$) where PHH frequency was higher for non-remitters than for remitters ($p<0.05$).

Supporting the clinical utility of these measures, we generated ROC curves and calculated the threshold for each parameter that distinguished remitters from non-remitters. All studied PHH parameters were able to predict the remission status of a given patient (all p-values$<0.05$) with PHH/Hyperglycemia duration ratio being the most sensitive parameter (ratio$<0.02$; Se = 86% and Sp = 68%) and PHH duration mean the most specific parameter (duration mean $<132$ min; Se = 76% and Sp = 74%).

Finally, investigating the correlations between the partial remission intensity (i.e., $IDAA_{1C}$ score) and PHH parameters, $IDAA_{1C}$ correlated moderately with PHH frequency ($R^2 = 0.10$; $p<0.001$) and strongly with $PHH_{AUC}$ ($R^2 = 0.32$; $p<0.001$) (**Table 2**). Notably, large variability in the PHH frequency was observed for intermediate $IDAA_{1C}$ values (i.e., in 8–10 range).

**Table 2. Linear mixed-models determination coefficients between parameters of PHH and glucose homeostasis.**

| | PHH frequency | | $PHH_{AUC}$ | | PHH duration mean | | PHH/Hyperglycemia duration ratio | | PHH/Hypoglycemia frequency ratio | |
|---|---|---|---|---|---|---|---|---|---|---|
| | $R^2$ | p-values | $R^2$ | p-values | $R^2$ | p-values | $R^2$ | p-values | $R^2$ | p-values |
| **CV** | 0.24 | $9.4e^{-10}$ | 0.22 | $5.82e^{-8}$ | 0.20 | $2.9e^{-7}$ | 0.50 | $2.2e^{-16}$ | 0.001 | 0.67 |
| **$IDAA_{1c}$** | 0.10 | $2.3e^{-5}$ | 0.32 | $4.04e^{-12}$ | 0.34 | $1.01e^{-11}$ | 0.30 | $4.97e^{-13}$ | 0.10 | $1.54e^{-4}$ |
| **MODD** | 0.05 | $1.2e^{-3}$ | 0.43 | $2.2e^{-16}$ | 0.40 | $3.6e^{-16}$ | 0.45 | $2.2e^{-16}$ | 0.17 | $4.47e^{-7}$ |
| **$Hypoglycemia_{<60}$ frequency** | 0.19 | $1.5e^{-10}$ | 0.02 | 0.07 | 0.03 | $3.7e^{-2}$ | 0.07 | $2.4e^{-4}$ | 0.12 | $7.88e^{-6}$ |
| **$TIT_{63-140}$** | 0.04 | $4.4e^{-3}$ | 0.40 | $2.2e^{-16}$ | 0.38 | $2.2e^{-16}$ | 0.30 | $2.2e^{-16}$ | 0.27 | $1.60e^{-11}$ |
| **$TIR_{70-180}$** | 0.06 | $9.3e^{-5}$ | 0.35 | $9.32e^{-16}$ | 0.35 | $6.69e^{-16}$ | 0.40 | $2.2e^{-16}$ | 0.18 | $3.74e^{-8}$ |
| **$TAR_{>180}$** | 0.01 | 0.06 | 0.42 | $2.2e^{-16}$ | 0.41 | $2.2e^{-16}$ | 0.28 | $3.74e^{-15}$ | 0.24 | $3.51e^{-10}$ |
| **$CPEP_{EST}$[#]** | 0.08 | $6.3e^{-3}$ | 0.17 | $2.68\ e^{-4}$ | 0.15 | $6.6e^{-4}$ | 0.29 | $9.56e^{-9}$ | 0.03 | 0.13 |

Marginal R-squared (coefficient of determination) were calculated using generalized linear mixed models. Results were considered as significant when p-value was under 0.05. Glycemic homeostasis markers and PHH parameters were evaluated at +3, +6, +9, +12 months after diagnosis. PHH, post-hypoglycemia hyperglycemia; Frequency, quantity of PHH per day; $PHH_{AUC}$, area under the curve of PHH; PHH/Hyperglycemia duration ratio, ratio between the total duration of PHH and the total duration of hyperglycemia; PHH/Hypoglycemia frequency ratio, ratio between daily frequency of PHH and daily frequency of hypoglycemia; $IDAA_{1C}$, Insulin Dose-Adjusted A1C = HbA1C + 4 x insulin doses/day/kg; CV, coefficient of variation for glucose; MODD, mean of daily differences; $TIR_{70-180}$, Time in range; $TIT_{63-140}$, Time in target; $TAR_{>180}$, Time above the range; $CPEP_{EST}$, estimation of c-peptide secretion calculated as described in Wentworth *et al* [25].

## High residual C-peptide secretion is associated with a reduction of PHH parameters

As PHH parameters overlapped for intermediate $IDAA_{1C}$ scores, we evaluated whether replacing this score with a residual β-cell secretion marker (i.e., $CPEP_{EST}$) might improve differences in PHH parameters among both remission groups. The patients were classified into four groups according to their residual estimated C-peptide secretion [31]: high (c-peptide>0.4 pmol/mL), intermediate (0.2 pmol/mL<c-peptide⩽0.4 pmol/mL), low (0.17 pmol/mL<c-peptide⩽0.2 pmol/mL), and undetectable (c- peptide⩽0.17 pmol/mL).

$CPEP_{EST}$ weakly correlated with PHH frequency ($R^2 = 0.08$; p<0.001), moderately with $PHH_{AUC}$ ($R^2 = 0.17$; p<0.001) and strongly with PHH/Hyperglycemia duration ratio ($R^2 = 0.29$; p<0.001). In addition, no correlation was found between residual $CPEP_{EST}$ secretion and PHH/Hypoglycemia frequency ratio ($R^2 = 0.03$; p = 0.13). Interestingly, patients with high residual secretion exhibited fewer and shorter PHH events, contrasting with patients with low or undetectable residual secretion (p<0.01) (**Table 2**). Finally, we observed that patients with intermediate secretion tended to have high intra-group variability in PHH frequency.

## PHH parameters correlate with glycemic homeostasis markers

As correlations between PHH parameters and β-cell residual secretion were moderate, we investigated whether PHH parameters would better correlate with continuous glucose monitoring metrics including glycemic variability (i.e., CV, mean of daily differences [MODD]) and time spent within different glycemic ranges (i.e., TIR70-180, TAR>180, hypoglycemia frequency [<60mg/dL]) (**Table 2**).

CV moderately-to-strongly correlated with nearly all PHH parameters ($p < 0.05$) except for PHH/hypoglycemia frequency ratio ($p > 0.05$) (**Table 2**). Interestingly, patients with CV values >36% exhibited three times more PHH events than patients with CV <36%. Also, MODD showed the best correlations among all glycemic homeostasis markers with all PHH parameters ($R^2 > 0.17$), except for PHH frequency. TIR70-180 inversely correlated with PHH parameters. Interestingly, TAR>180 weakly correlated with PHH frequency. Also, the hypoglycemia frequency was the best correlation with PHH frequency ($R^2 = 0.19$; $p < 0.001$) (**Fig 2**).

Focusing on hypoglycemia and glycemic variability, patients were categorized into three different groups that demonstrated specific combinations of glycemic variability (i.e., CV) and PHH parameters (i.e., $PHH_{AUC}$ and PHH/Hypoglycemia frequency ratio) (**Fig 3**). Group 1 corresponds to patients with stable type 1 diabetes (i.e., CV <36%), low PHH/Hypoglycemia frequency ratio (<0.25), and very low $PHH_{AUC}$ (<25000). Conversely, patients in Group 2 demonstrate a highly variable PHH/Hypoglycemia frequency ratio (0.1–0.5) with low-to-intermediate $PHH_{AUC}$ (<60000) and high glycemic variability (CV>36%). Finally, patients in Group 3 have very rare hypoglycemia that is often followed by a PHH (PHH/Hypoglycemia frequency ratio [>0.4]) of high amplitude ($PHH_{AUC}$ [>60000]). Interestingly, group 1 was mostly composed of remitters, and groups 2 and 3 of non-remitters. Finally, the temporal evolution of each patient revealed that more than half of the patients stayed in the same group (53%) during the first year of type 1 diabetes while others switched from one group to another except from Groups 2 or 3 to Group 1.

## Patient glucotypes identify patients at risk for PHH

To better characterize patients with non-discriminative indexes of glycemic homeostasis markers (i.e., intermediate $CPEP_{EST}$ secretions and $IDAA_{1C}$ scores), we studied PHH parameter distribution in the four distinct glucotypes that were previously described by our team in patients during the first year after type 1 diabetes onset [21].

Glucotypes 1 and 2 demonstrated very low PHH duration means corresponding to a low PHH/Hyperglycemia duration ratio (< 0.01). While $PHH_{AUC}$ of glucotype 3 remained similar to glucotype 2, we observed a major increase in PHH frequency and a concomitant rise in PHH/Hyperglycemia duration ratio in glucotype 3. Finally, contrasting with other glucotypes, glucotype 4 demonstrated the highest PHH/Hyperglycemia duration ratio (0.1 ± 0.08). Notably, glucotypes 2 and 4 were characterized by a high PHH/Hypoglycemia frequency ratio though high variability could be observed between the patients (i.e., respectively 0.61 ± 0.23 and 0.77 ± 0.38) (**Fig 4**).

Circadian PHH analysis showed an increase of PHH frequency in glucotypes 3 and 4 with the highest differences being observed during the $day_{10-4}$ and $evening_{4-10}$ ($p < 0.05$). Interestingly, PHH frequency in the $morning_{5-10}$ did not differ across glucotypes ($p > 0.05$). Moreover, the largest $PHH_{AUC}$ were observed during the $night_{10-5}$ ($p < 0.001$) regardless of glucotypes with patients in glucotype 1 experiencing the smallest $PHH_{AUC}$.

## Discussion

Partial remission reflects a transient recovery of β-cell function with increased insulin secretion [31] and improved peripheral insulin sensitivity [32], leading to decreased dependence on exogenous insulin and optimal glycemic control (e.g., TIR70-180, glycemia variability) [1]. When β-cell function further declines, insulin requirements and glycemic variability increase, corresponding to the end of partial remission and an increased hypoglycemic risk. Current definitions and biomarkers used to identify partial remission (i.e., residual secretion [33,34] or $IDAA_{1C}$ score [24]) either require invasive blood sampling or present several limitations to describe the evolution of glycemic homeostasis (e.g., hypoglycemia, glycemic variability,

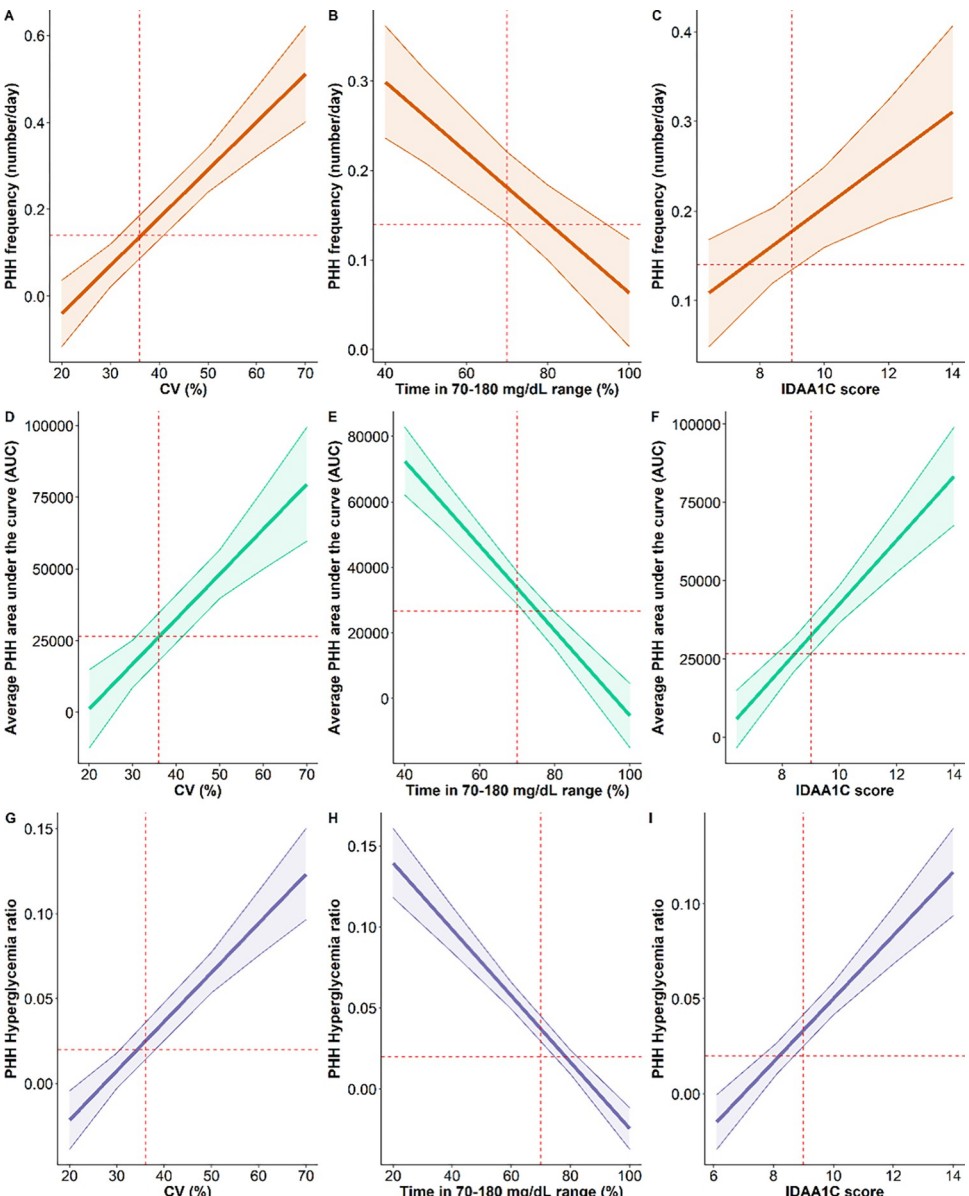

**Fig 2. Determination coefficient of PHH frequency (A-C), PHH$_{AUC}$ (D-F) and PHH/Hyperglycemia duration ratio (G-I) according to glycemic homeostasis markers.** Parameters of PHH parameters and glycemic homeostasis markers were obtained at +3, +6, +9, and +12 months after the diagnosis. Marginal R-squared (coefficient of determination) were calculated using generalized linear mixed models. The vertical *red dashed lines* represent specific thresholds of glucose homeostasis parameters (CV = 36%, time in 70–180 mg/dL range = 70% and IDAA$_{1C}$ score = 9). The horizontal *red dashed lines* represent specific status-related thresholds of PHH parameters (PHH frequency = 0.14, PHH$_{AUC}$ = 26567 and PHH/Hyperglycemia duration ratio = 0.02). Panels **A-C** represent the regression results with 95% CI bands (*shaded zone*) between PHH frequency and CV (A), TIR (B) and IDAA$_{1C}$ (C). Panels **D-F** represent the regression results with 95% CI bands (*shaded zone*) between PHH$_{AUC}$ and CV (D), time in 70–180 mg/dL range (E) and IDAA$_{1C}$ (F). Panels **G-I** represent the regression results with 95% CI bands (*shaded zone*) between PHH frequency and CV (G), time in 70–180 mg/dL range (H) and IDAA$_{1C}$ (I). Correlation coefficients (R) are shown according to the PHH frequency (**A-C**), PHHAUC (**D-F**) and PHH/Hyperglycemia duration ratio (**G-I**); Abbreviations: PHH frequency (number/day), PHH frequency; Average PHH area under the curve (AUC), PHH$_{AUC}$; PHH Hyperglycemia ratio, PHH/Hyperglycemia duration ratio; CV (%), coefficient of variation for glucose; Time in 70–180 mg/dL range, Time in range; IDAA1C score, Insulin Dose-Adjusted A1C = HbA$_{1C}$ + 4 x insulin doses/day/kg. The level of significance of the correlations is represented after the *regression coefficient* as follows: Nonsignificant (*ns*), p<0.05 (*), p<0.01 (**), p<0.001 (***).

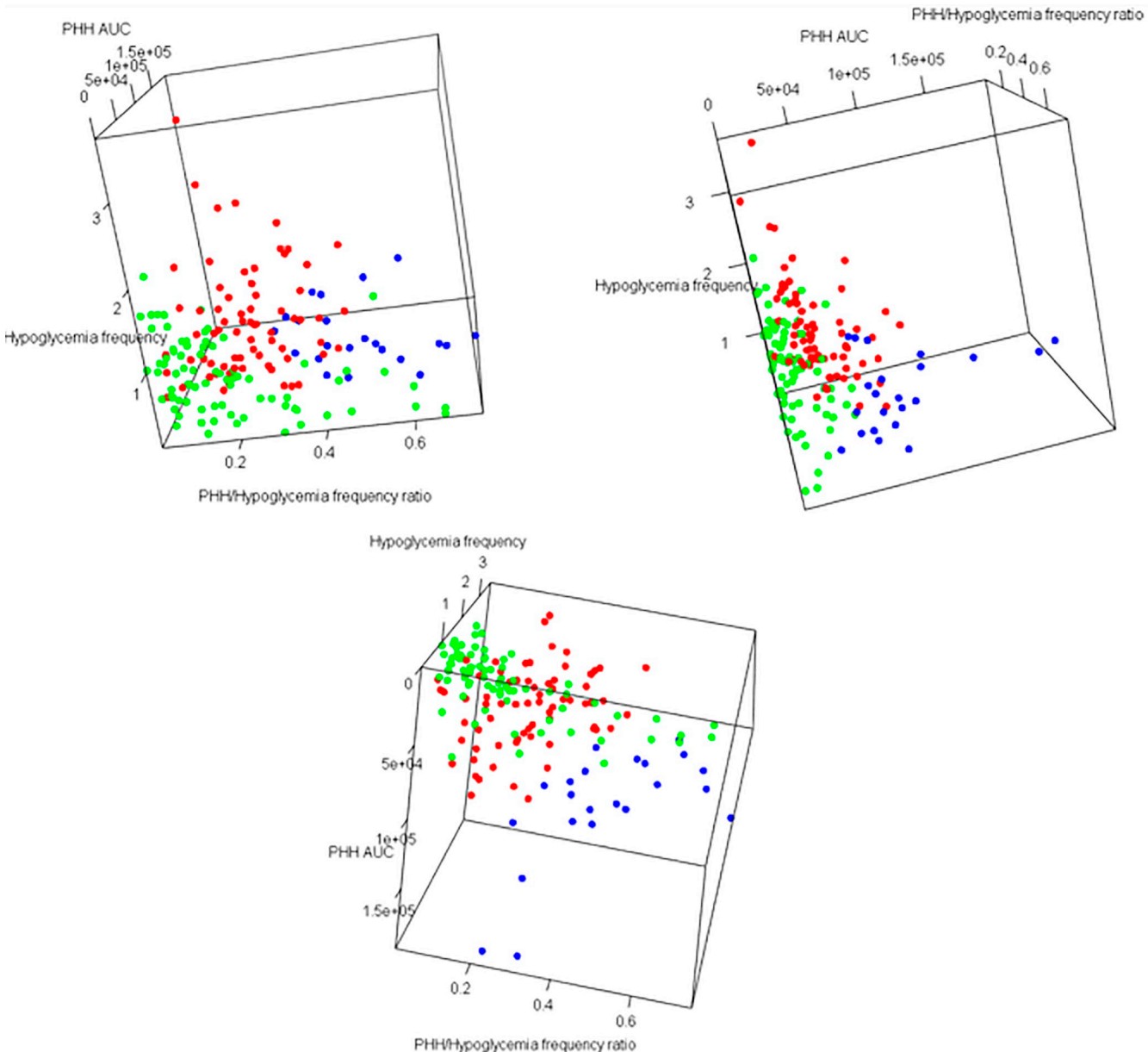

**Fig 3. Multidimensional representation of PHH parameters according to the three target patient groups identified using hierarchical clustering.** PHH parameters and glycemic homeostasis markers were obtained at +3, +6, +9 and +12 months after the diagnosis. Three-dimensional representation of PHH$_{AUC}$, frequency of hypoglycemia and PHH Hypo ratio. Each color of the dots is specific to a target group (green = Group 1, red = Group 2, blue = Group 3).

insulin sensitivity) [20]. In this context, there is a need for minimal-invasive reliable markers that allow the characterization and stratification of patients with type 1 diabetes based on their glycemic status.

In our GLUREDIA study, we extensively characterized PHH, a new marker of glycemic variability, in new-onset type 1 diabetes patients. Using 12-week continuous glucose monitoring metrics, we identified PHH parameters (e.g., PHH duration mean and PHH/hyperglycemia duration ratio) as both highly sensitive and specific markers to differentiate between patients undergoing remission or not. Furthermore, we established clinically relevant

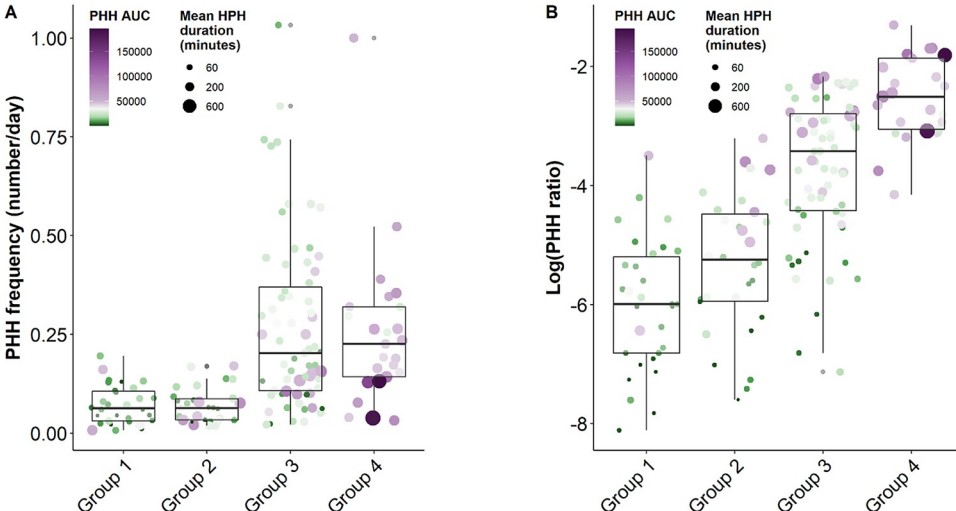

**Fig 4. Distribution of PHH parameters according to the glucotypes [21].** PHH parameters and glycemic homeostasis markers were obtained at +3, +6, +9 and +12 months after the diagnosis. The glucotypes were determined as described by our group [21]. **(A)** Boxplots of PHH frequency according to glucotypes (group 1–4). **(B)** Boxplot of PHH/hyperglycemia duration ratio according to glucotypes (group 1–4). Size of the dots corresponds to the PHH duration mean. Color of the dots represents levels of $PHH_{AUC}$, using continuous colored gradient scale (*green-purple*) with $PHH_{AUC}$ = 26567 in *white*. Abbreviations: PHH frequency, quantity of PHH per day; PHH AUC, area under the curve of PHH; Mean HPH duration, average duration of a PHH; Log (PHH ratio), logarithm function of ratio between the total duration of PHH and the total duration of hyperglycemia.

segregation thresholds for each of them. Also, most PHH parameters demonstrated high variability across the day being the highest during the day$_{10-4}$ and evening$_{4-10}$. This suggested that reduced residual β-cell function in combination with behavioral habits might be major triggers of PHH events. Finally, PHH parameters strongly correlated with continuous glucose monitoring metrics (i.e., TIR70-180, TAR>180, CV) while correlations with residual β- cell secretion were only weak to moderate.

Integrating previously described glucotypes [21], we showed that PHH parameters mirrored the progressive increase of glycemia variability across the various glucotypes (**Fig 4**). Interestingly, refining the ROC curves analysis, most false positive and negative patients had an IDAA$_{1C}$ score in the 8–10 range though distinctively distributing across glucotypes 2 and 3 that were previously described by our team [21]. This observation independently validates our previous results, further confirms PHH parameters as reliable markers of glycemic homeostasis and supports recent evidence that partial remission should be considered as a continuum rather than a dichotomic phenomenon [21].

As previously mentioned, glycemic variability and hyperglycemia are both independent predictors of micro and macro-vascular complications through protein glycation (AGE) and oxidative stress [35,36]. Paradoxically, continuous glucose monitoring metrics are not integrated into the main scores defining partial remission. In our study, most PHH parameters demonstrated to be strong markers of continuous glucose monitoring metrics (e.g., CV and MODD) (**Fig 2**/**Table 2**). Going further, our results support the clinical validity of the diabetes stability definition based on CV as a steeper increase of PHH parameters values was observed from the value of 36% [37]. Moreover, Cerriello et al. showed that PHH occurrence was an independent risk factor of oxidative stress. Indeed, patients with type 1 diabetes presenting PHH displayed a pejoration of endothelial function, an increase in inflammation and an increase of oxidative stress markers in an ischemic-reperfusion-like effect [18]. These findings

together reinforce PHH parameters as reliable and independent markers of glycemic homeostasis and a potential predictor of type 1 diabetes-related chronic complications.

Carbohydrate intake and management of hypoglycemia influence glycemic control in a patient-dependent way [38]. Indeed, in our study, PHH principally occurred during the $day_{10\text{-}4}$ and $evening_{4\text{-}10}$ when the parents and/or the child are active and may strongly influence the glycemic control in some avoidable. Following this idea, the combination of PHH parameters ($PHH_{AUC}$ and PHH/Hypoglycemia frequency ratio) and hypoglycemia frequency subdivided our type 1 diabetes patient cohort into three subgroups. Among these groups, the majority of patients in Group 2 frequently overtreated their hypoglycemia when trying to reach a normoglycemic state resulting in short and frequent PHH, and consequently high glycemic variability. Therefore, therapeutic education focusing on the management of hypoglycemia [8] and targeting a PHH/Hypoglycemia frequency ratio below 0.25 may considerably improve disease control in these patients, by preventing further PHH events with reduced glycemic variability.

The principal strength of our multicentric pediatric study relies on the cross-sectional data analysis that integrates complementary markers of glycemic homeostasis during the first year of type 1 diabetes. In addition, PHH parameters thresholds allow an easy, reliable and poorly-invasive determination of the remission status and diabetes control using a freely-available algorithm.

Our study was also limited by the cross-sectional analysis of all three parameters (i.e., clinic, secretion and continuous glucose monitoring metrics) that were only available for a subset of patients (i.e., 70%). Also, the sensor manufacturer may influence the data though most of our dataset (i.e., >90%) was obtained from Freestyle Libre$^{®}$ though no sensor-related differences were identified on principal component analysis [21]. Furthermore, the small number of patients under pump delivery system did not allow us to perform subanalysis according to the insulin regimen (i.e., multiple daily injections *vs* pump). Moreover, data including the type of rapid insulin analog were not collected and thus could not be analyzed.

Finally, we believe it would also be useful to evaluate these clinical parameters in another, larger cohort of patients with type 1 diabetes to confirm these results. Indeed, these parameters could be studied in cohorts of adult patients or patients with long-term diabetes, for example.

## Conclusion

In conclusion, our study provides a user-friendly software that automatically identifies and characterizes PHH glycemic patterns on CGM data in patients with new-onset type 1 diabetes. Parameters of PHH demonstrated strong correlations with routine markers of glucose homeostasis (e.g., $TIR_{70\text{-}180}$, $TAR_{>180}$) and glycemic variability (e.g., CV), but only moderate correlation with residual β-cell secretion estimates. These parameters distinguished remitters from non-remitters (e.g., PHH/Hyperglycemia duration ratio and $PHH_{AUC}$), supporting PHH as new minimal invasive markers of PR. We believe that integrating PHH parameters in continuous glucose monitoring reports may rise awareness on hypoglycemia and foster patient-specific therapeutic interventions (e.g., management of carbohydrate intake).

## Author Contributions

**Conceptualization:** Antoine A. Harvengt, Olivier G. Polle, Philippe A. Lysy.

**Formal analysis:** Manon Martin, Aline van Maanen, Laurent Gatto.

**Funding acquisition:** Philippe A. Lysy.

**Investigation:** Antoine A. Harvengt, Olivier G. Polle, Philippe A. Lysy.

**Methodology:** Antoine A. Harvengt, Olivier G. Polle, Manon Martin, Aline van Maanen, Laurent Gatto, Philippe A. Lysy.

**Supervision:** Philippe A. Lysy.

**Writing – original draft:** Antoine A. Harvengt, Olivier G. Polle, Philippe A. Lysy.

**Writing – review & editing:** Antoine A. Harvengt, Olivier G. Polle, Philippe A. Lysy.

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
