## [Decision Letter · Decision Letter 0]

28 Mar 2023

PONE-D-22-33441Post-Hypoglycemic Hyperglycemia Are Highly Relevant Markers For Stratification Of Glycemic Variability and Remission Status Of Pediatric Patients With New-Onset Type 1 Diabetes.PLOS ONE

Dear Dr. Harvengt,

Thank you for submitting your manuscript to PLOS ONE. After careful consideration, we feel that it has merit but does not fully meet PLOS ONE’s publication criteria as it currently stands. Therefore, we invite you to submit a revised version of the manuscript that addresses the points raised during the review process.Authors should indicate as part of publication requirement, how the current study relate and different from EHPHICA, DIATAG, and GLUREDIA studies to bring distinction.  Authors should indicate and or address any conflict of interest between these studies(EHPHICA, DIATAG, and GLUREDIA)Authors should define all abbreviations upon first time usage, authors should provide some background/rationale in the abstract section to put the study into perspective to help readers.Study design, study site and study population should be well defined and explainedthe main rationale of the study was rather stated within the first three paragraphs of the discussion instead of the introduction, authors should revise as such.For discussions, authors should state their results compare with related studies and state reasons for observed differences or similaritiesPlease submit your revised manuscript by May 12 2023 11:59PM. If you will need more time than this to complete your revisions, please reply to this message or contact the journal office at plosone@plos.org. Please include the following items when submitting your revised manuscript:A rebuttal letter that responds to each point raised by the academic editor and reviewer(s). You should upload this letter as a separate file labeled 'Response to Reviewers'.A marked-up copy of your manuscript that highlights changes made to the original version. You should upload this as a separate file labeled 'Revised Manuscript with Track Changes'.An unmarked version of your revised paper without tracked changes. You should upload this as a separate file labeled 'Manuscript'.

We look forward to receiving your revised manuscript.

Kind regards,

Samuel Asamoah Sakyi, Ph.D

Academic Editor

PLOS ONE

Journal Requirements:

“Grant funding from Leona M. and Harry B. Helmsley Charitable Trust.”

Reviewers' comments:

Reviewer's Responses to Questions

**Comments to the Author**

1. Is the manuscript technically sound, and do the data support the conclusions?

Reviewer #1: No

Reviewer #2: Yes

2. Has the statistical analysis been performed appropriately and rigorously? 

Reviewer #1: Yes

Reviewer #2: Yes

3. Have the authors made all data underlying the findings in their manuscript fully available?

Reviewer #1: Yes

Reviewer #2: Yes

4. Is the manuscript presented in an intelligible fashion and written in standard English?

Reviewer #1: No

Reviewer #2: Yes

5. Review Comments to the Author

Reviewer #1: Abstract

The abstract should be restructured with the results section revised to include means percentages and standard deviatiions

What was the design of the study?

The AUTHORS SHOULD INDICATE HOW THE GLYCATED hB was estimated

What is the deficnation of glucotypes and how did the authors come by the types

The limitation should be added to the main text of the discussion

The length of the conclusion should also be reduced

Reviewer #2: General Comment

This manuscript is a well thought of experiment that evaluates whether parameters of post-hypoglycemic hyperglycemia (PHH) correlated with glycemic homeostasis during the first year after type 1 diabetes onset and tries to use this to distinguish pediatric patients undergoing partial remission or not. However there are some specific comments that need attention

Abstract;

There are a number of grammatical errors that authors need to correct. Moreover, there are abbreviations that need to be defined at first mention. Under the results, authors need to quote the estimates that indicate these PHH parameters are able to discriminate between remitters and non-remitters.

Methodology

Under continuous glucose monitoring, authors indicate varied devices being used for this purpose but fail to indicate how these were standardized for the purpose of this study. This section requires further proof reading (see attached reviewer pdf). Authors do not justify the sample size used in the study

Results

Authors should re-check the total number of outpatient visits in the first paragraph. There are sections under frequency of PHH which are better suited for methodology or discussion (see attached reviewer pdf). Authors base conclusions on R2 values that are minimal with statistical significance and must discuss with this in mind. Authors must thus be cautious of their conclusions on these estimates.

Discussion

The first paragraph under the discussion is more of a literature for background than the beginning of discussion which usually has the major findings from the results. The inclusion criteria of experience with COVID-19 need further information as the frequency and how recent are likely to influence responses

Conclusion

The Conclusion should specifically speak to aims of the study followed by implications of these findings.

6. PLOS authors have the option to publish the peer review history of their article (what does this mean?). If published, this will include your full peer review and any attached files.

Reviewer #1: **Yes: **RICHARD KD EPHRAIM

Reviewer #2: No

---

## [Author Response · Author response to Decision Letter 0]

19 Apr 2023

Cfr "Response to reviewers" in download document

---

## [Editor Report · Decision Letter 1]

7 May 2023

PONE-D-22-33441R1Post-Hypoglycemic Hyperglycemia Are Highly Relevant Markers For Stratification Of Glycemic Variability and Remission Status Of Pediatric Patients With New-Onset Type 1 Diabetes.PLOS ONE

Dear Dr. Lysy,

Thank you for submitting your manuscript to PLOS ONE. After careful consideration, we feel that it has merit but does not fully meet PLOS ONE’s publication criteria as it currently stands. Therefore, we invite you to submit a revised version of the manuscript that addresses the points raised during the review process.

Reviewers are of the view that you have responded adequately to their concern, but there still few MINOR CORRECTION.

We look forward to receiving your revised manuscript.

Kind regards,

Samuel Asamoah Sakyi, Ph.D

Academic Editor

PLOS ONE

Journal Requirements:

Additional Editor Comments:

Authors has responded adequately to most of the concern raised by the reviewers, however, some few outstanding concern,

1. the revised manuscript were not numbered, and this make it difficult to relate the responses to the changes, authors should number the pages.

2. Authors should not assume everybody has read their previous study, the current manuscript should be able to make meaning to it readers without recourse to their earlier paper, authors should factor this in their write up.

3. still first-time usage of most acronyms are not defined

4. table 1 what do authors mean by global ??? do they mean total??
---

## [Author Response · Author response to Decision Letter 1]

11 May 2023

Cfr "Response to editor" in download document

---

## [Decision Letter · Decision Letter 2]

18 Jul 2023

PONE-D-22-33441R2Post-Hypoglycemic Hyperglycemia Are Highly Relevant Markers For Stratification Of Glycemic Variability and Remission Status Of Pediatric Patients With New-Onset Type 1 Diabetes.PLOS ONE

Dear Dr. Antoine Harvengt

Thank you for submitting your manuscript to PLOS ONE. After careful consideration, we feel that it has merit but does not fully meet PLOS ONE’s publication criteria as it currently stands. Therefore, we invite you to submit a revised version of the manuscript that addresses the points raised during the review process.

We look forward to receiving your revised manuscript.

Kind regards,

Engidaw Fentahun Enyew, MSc

Academic Editor

PLOS ONE

Reviewers' comments:

Reviewer's Responses to Questions

**Comments to the Author**

1. If the authors have adequately addressed your comments raised in a previous round of review and you feel that this manuscript is now acceptable for publication, you may indicate that here to bypass the “Comments to the Author” section, enter your conflict of interest statement in the “Confidential to Editor” section, and submit your "Accept" recommendation.

Reviewer #3: (No Response)

Reviewer #4: (No Response)

2. Is the manuscript technically sound, and do the data support the conclusions?

Reviewer #3: Yes

Reviewer #4: Partly

3. Has the statistical analysis been performed appropriately and rigorously? 

Reviewer #3: Yes

Reviewer #4: I Don't Know

4. Have the authors made all data underlying the findings in their manuscript fully available?

Reviewer #3: Yes

Reviewer #4: No

5. Is the manuscript presented in an intelligible fashion and written in standard English?

Reviewer #3: Yes

Reviewer #4: Yes

6. Review Comments to the Author

Reviewer #3: Remarks to the authors

Manuscript Number: PONE-D-22-33441R2

Summary of the research

This study is interestingly addressing the partial remission occurred in the first year after the onset of type 1 Diabetes in pediatric patients. The fluctuations in blood glucose levels have unpleasant outcomes, and greatly affect the quality of life for the kids and their parents. Thus, I think the prediction of the glycemic variability using less invasive approach would contribute to highlight the best practices of therapeutic managements.

comments

Title

The title is descriptive, concise, and interesting, but it would be more accurate if the partial remission was addressed.

Abstract

The abstract is simple and specific, and well structured.

The "CGM" abbreviation should be defined at the first mention in the abstract. manuscript (i.e. CGM, continuous glucose monitoring)

Introduction

The authors start with a general overview of Type 1 diabetes, and the concerns over the incidences of glycemic variability in the pediatric patients. The research question was introduced through reviewing the most recent literature. The authors also stated the drawbacks of other markers used to address the hypoglycemic and hyperglycemic events (i.e. β-cell function and Insulin-Dose Adjusted A1C).

Finally, the key characteristics and the aim of the study were outlined.

Methodology

The authors stated the criteria of participants' enrollment in the study, and they also have considered the ethical basis of their work. All the methods, data collection and procedures were clearly described.

In the "partial remission" section, why did you only depend on IDAA1C in the classification of remitters and non-remiiters according to reference #24?

A recently published research (2022) have pointed out the limitation of this criteria, which was published in 2009 study. So, please refer to "Nwosu BU. Partial Clinical Remission of Type 1 Diabetes: The Need for an Integrated Functional Definition Based on Insulin-Dose Adjusted A1c and Insulin Sensitivity Score. Front Endocrinol (Lausanne). 2022 May 3;13:884219. doi: 10.3389/fendo.2022.884219. PMID: 35592786; PMCID: PMC9110823"

Results

The findings of research questions were clearly presented in the results section. However, I have noticed that these results were more presented in tables and visualized as figures according to the participant's status classification (i.e. remitters versus non-remitters), rather than the four glucotypes. Is there a specific explanation for that?

Discussion and Conclusion

The authors did highlight the findings of the study and compare the results between the main glucotypes under investigation. They have described the strengths of the study as well as the limitations. However, they did not introduce suggestions to overcome these limitations for PHH markers as a potential markers of PR.

Finally, Do the authors suggest any future studies that need to be carried out?

References

The References listed are relevant and mostly are recently published.

The references (11, 17, 19 and 23) should be in the same format.

Figures

I think the figures need improvements in terms of font size of the labels, resolution, and the thickness of lines and ticks.

Tables

The Data summarized in (table 1) should be rounded appropriately, the decimal places should be presented in a uniform pattern.

Reviewer #4: 1. Summary of the research

The main research question is clear and well-defined, that is to asses the correlation, if any, between PHH parameters and glucose homeostasis in newly diagnosed T1D pediatric patients and differentiate remitters from nonremitters using least invasive technique, which is another noble goal.

Except for some undefined abbreviations, the introduction is clear, concise and informative and gives a good background to the audience about the current subject.

As for the study design I found it ambiguous in many points as in the true number of recruiters and then the number of CGM metrics which represents the sample size, Plus referring to a previous study for the reader to visit and extract the details. Or even mentioning a certain detail that does not match the previous study which referred to as the mother study. This was misleading and exhausting indeed.

The results are described in detail using clear language and is presented by more than one form (tables and graphs). Figure 1 is really fascinating as it unravels what words could not clarify in the manuscript. However, although I am not a statistics expert, sometimes the deduction does not match the data presented by the figure.

An edited form of the manuscript is attached to view all the review point and comments but below are some examples of which.

2. Examples and evidence

Major issues:

a. In the abstract, methods section, first line: what does GLUREDIA abbreviate? If it has a full term, please mention it first and if not, I suggest defining it shortly before mentioning it or just say: "in the current study, named GLUREDIA, ..." or " In this study, called GLUREDIA ...."This is because mentioning it in the abstract preceded by the article "the" gives the impression that it is a well-defined famous-in-the-field abbreviation, which is not true and hence misleading to the audience

b. The same for " EPHICA " in the introduction fourth paragraph, 3rd line and "DIATAG" in the study design section first paragraph, 1st line.

c. The repeated referring to the authors' previous works with hiding some important details to force the reader to go back to these previous studies is uncomfortable, giving the impression it is not a standalone study, besides revealing some controversies with those studies like in the study design the authors mentioned that the current study is subsidiary from the "DIATAG". It is also mentioned that the age upper limit is 17 years. Returning to the DIATAG study in reference 21, the upper limit for age was 18 not 17. In the current study, is the upper limit for age 18 like in the DIATAG and this is a typo? or there is a reason for excluding those aged above 17. please specify it. So, please mention the exact age range in this section.

d. Upon visiting the registered clinical trial (NCT04007809) to which this study belongs the true number of recruited participants was 98, What were the reasons for excluding these 27 participants? Please mention the exact number of participants clearly in this section.

e. The exclusion criteria also were found at the registered clinical trial (NCT04007809) not in reference 21 as mentioned in the manuscripts.

So, from the abovementioned (c, d & e) if the authors had decided to include only 71 participant from the 98, by selecting them randomly before the study began, and age range up to 17 only not 18, with no convincing reason, I suggest mentioning all the study details, conditions, criteria in the methodology section, to present the current study as an independent one without referring to previous works or clinical trials.

f. In the introduction fourth paragraph, 3rd line defining the PHH as hypoglycemia followed by hyperglycemia does not sound good. because English-wise the term refers to hyperglycemia that is described by being post hypoglycemic, i.e happens after hypoglycemia, (unless you mean to describe the whole phenomena, which will not be compatible with what I understood from Fig 1 )

the suggested rephrased definition is found in the attached document.

g. In the results section under "Clinical and anthropometric characteristics of the GLUREDIA cohort" the manipulation of the numbers is not clearly understood. In other words, there were no justifications for the decrease in number in the first step:

- How does quarterly visits in a year duration for 71 patients count for 244? i.e: 71*4= 284, not 244, so you should mention that there were 40 missing visits, before mentioning the exclusion of 52 visits from the 244 that gives 192 , the supposed (192) visits that corresponds to 66 patients (provided not every patient from the 66 attended all the 4 visits), was then after under the title "PHH is frequent during the first year……..etc." seen as (194) CGM metrics. Then, it appears as (193) glucose records under the title " Partial remission is associated with fewer and lower PHH occurrence ". Again, in table 1 n=194. So, what is the right number 194, 192 or 193?

- I think this is a critical issue that affect all the numbers in calculations and statistics.

h. Where would a reader find all the data which the authors have referred to as " fully available without restriction, in any part of the manuscript, or in another public repository or as a supplementary material?

i. In the results which hyperglycemia is meant in (PHH/Hyperglycemia duration ratio)?

Minor issues:

The minor issues are included within the review comments on the attached word doc of reviewed manuscript

7. PLOS authors have the option to publish the peer review history of their article (what does this mean?). If published, this will include your full peer review and any attached files.

Reviewer #3: No

Reviewer #4: No

---

## [Decision Letter · Decision Letter 3]

14 Nov 2023

Post-Hypoglycemic Hyperglycemia Are Highly Relevant Markers For Stratification Of Glycemic Variability and Partial Remission Status Of Pediatric Patients With New-Onset Type 1 Diabetes.

PONE-D-22-33441R3

Dear Dr. Harvengt,

We’re pleased to inform you that your manuscript has been judged scientifically suitable for publication and will be formally accepted for publication once it meets all outstanding technical requirements.

Kind regards,

Aleksandra Klisic

Academic Editor

PLOS ONE

Additional Editor Comments (optional):

Reviewers' comments:

Reviewer's Responses to Questions

**Comments to the Author**

1. If the authors have adequately addressed your comments raised in a previous round of review and you feel that this manuscript is now acceptable for publication, you may indicate that here to bypass the “Comments to the Author” section, enter your conflict of interest statement in the “Confidential to Editor” section, and submit your "Accept" recommendation.

Reviewer #2: All comments have been addressed

Reviewer #3: All comments have been addressed

2. Is the manuscript technically sound, and do the data support the conclusions?

Reviewer #2: Yes

Reviewer #3: Yes

3. Has the statistical analysis been performed appropriately and rigorously? 

Reviewer #2: Yes

Reviewer #3: Yes

4. Have the authors made all data underlying the findings in their manuscript fully available?

Reviewer #2: Yes

Reviewer #3: Yes

5. Is the manuscript presented in an intelligible fashion and written in standard English?

Reviewer #2: Yes

Reviewer #3: Yes

6. Review Comments to the Author

Reviewer #2: Authors have significantly responded to review comments in the current draft.

These were majorly on defining abbreviations and discussion based on estimates reported in the results section.

Reviewer #3: After reviewing the manuscript, I would like to thank the authors for addressing my initial comments. All the fore mentioned comments were accordingly explained or amended. I have no additional comments, but there is only a typographic error when representing the uppercase letters related to the "GLUREDIA" abbreviation in the abstract.

7. PLOS authors have the option to publish the peer review history of their article (what does this mean?). If published, this will include your full peer review and any attached files.

Reviewer #2: No

Reviewer #3: No

---

## [Editor Report · Acceptance letter]

22 Nov 2023

PONE-D-22-33441R3 

Post-Hypoglycemic Hyperglycemia Are Highly Relevant Markers For Stratification Of Glycemic Variability and Partial Remission Status Of Pediatric Patients With New-Onset Type 1 Diabetes. 

Dear Dr. Harvengt:

I'm pleased to inform you that your manuscript has been deemed suitable for publication in PLOS ONE. Congratulations! Your manuscript is now with our production department. 

Kind regards, 

on behalf of

Dr. Aleksandra Klisic 

Academic Editor

PLOS ONE